# Evidence for the formation of silicic lava by pyroclast sintering

Annabelle Foster [1] ✉, Fabian B. Wadsworth [1] ✉, Hugh Tuffen [2], Holly E. Unwin [2] & Madeleine C. S. Humphreys[1]

Silicic lavas can be produced by the sintering of pyroclasts in the volcanic sub-surface, and then advected out of the vent. Here, we provide evidence for this mechanism preserved in the exposed post-glacial remnants of a silicic volcanic conduit at Hrafntinnuhryggur, Krafla volcano, Iceland. We show that the conduit margins are a clast-supported pumice lapilli tuff deposit that grades continuously into dense obsidian and that the obsidian contains cuspate relict clast boundaries and country rock lithic fragments throughout. Transects of $H_2O$ concentrations across the conduit show that the magma was degassed to different degrees laterally with systematic spatial variation that is consistent with progressive conduit clogging and final gas pressurisation. Textures in the overlying effusive lavas record the variably sheared and brecciated remnant of the same in-conduit sintering. This record of a silicic conduit system connected to upper eruptive deposits provides support for the 'cryptic fragmentation model' for effusive silicic volcanism.

The 2008 eruption of Volcán Chaitén and the 2011-12 eruption of Cordón Caulle, both in Chile, have reframed our understanding of silicic volcanism. Direct observations from those iconic and hazardous events showed that silicic eruptions can exhibit simultaneous and sustained effusive lava production alongside explosive activity producing pyroclastic deposits[1–10]. Such sustained simultaneity of explosive and effusive dynamics at the same vent has challenged traditional conceptions of the shallow volcanic sub-surface and the physical processes that are inferred. The 'cryptic fragmentation model' explains this coincidence of lava effusion and energetic pyroclastic venting via the proposal that the lava is itself produced from pyroclasts by in-conduit sintering and re-amalgamation of fragmental magmatic material and lithics[11]. This model has been supported by textural evidence diagnostic of clastogenic sintering in all products of hybrid explosive-effusive silicic eruptions including fall deposits (i.e. the explosive component)[1,12–14], volcanic bombs (i.e. a Vulcanian explosive component)[1,4,15–17], and the lava itself (i.e. the apparently effusive component)[1,2,18]. While this model appears to be broadly consistent with the evolution from explosive to hybrid explosive-effusive, and finally to effusive eruption styles, there remain key questions about these transitions that can only be answered by detailed investigation of

field examples in which the shallow interiors of volcanoes can be observed directly.

Brittle fragmentation of continuous magma to form pyroclasts is the singular event that typifies explosive volcanism[19]. If the conditions for magma fragmentation are met, conduit flow models for silicic magma predict that fragmentation occurs at ≥500 m[20] depth and up to several kilometres[21] below the Earth's surface, depending on the model used and the conditions of magma ascent invoked. Therefore, there can exist an extensive transport pathway between fragmentation and the surface in which a turbulent granular dispersion of pyroclasts can remain hot and may stick to the conduit walls via viscous or inertial interactions[2,11–13,16]. To date and recently, this process has been largely inferred on the basis of compelling, but ultimately indirect evidence from pyroclasts erupted sub-aerially, which show signs of being sintered at conduit walls, before being plucked, and re-entrained in the eruption column[1,12].

In addition to initial pyroclast impact, capture, and sticking during explosive activity, the cryptic fragmentation model goes further and predicts that as a silicic eruption progresses, an approximately conduit-parallel sintering front results in the progressive aggradation of dense obsidian-like conduit margin material, undergoing time-

[1]Earth Science, Durham University, Science Labs, Durham DL1 3LE, UK. [2]Lancaster Environment Centre, Lancaster University, Lancaster LA1 4YQ, UK. ✉e-mail: annabelle.foster@durham.ac.uk; fabian.wadsworth@gmail.com

dependent densification[1,11,22]. As this process proceeds, it is thought that the conduit progressively occludes and eventually that the marginal deposits are advected out as lava. Despite the shallow sub-surface pyroclastic origins of the lava, the nature of sintering is such that the result often appears texturally indistinguishable from magma that ascends without having undergone fragmentation at all[23]. However, as with the initial phases of pyroclast sticking described above, much of the dynamics that underpin these model conceptions of silicic eruptions remain unvalidated by any direct field observations, beyond indirect evidence from tephra and bombs.

Here, we aim to test and develop these hypotheses directly by focussing on field evidence at a well preserved conduit margin. To do this, we examine a well-exposed example of silicic conduit-filling rocks and report relict textural and geochemical data that we interpret as evidence for their petrogenesis by sintering.

## Results and discussion

### A preserved silicic system at Hrafntinnuhryggur, Krafla volcano

Compared with subaerial deposits of volcanic eruptions, well-preserved and dissected volcanic conduits and/or vent systems are relatively rare worldwide. Key silicic examples include: the Mule Creek vent system in New Mexico, U.S.A.[22,24], and Rauðufossafjöll (specifically the Thumall and Skriðugil conduit exposures) on the edge of the Torfajokull volcano, Iceland[18,25]. These examples can be supplemented by observations made of either the partially dissected interior of volcanic domes and proximal lava areas[26], and, in some some cases, drill core sections into the sub-surface[11,27]. Despite these key and important examples, it appears particularly rare to find a field example in which exposed conduit feeder systems are stratigraphically associated with exposure of surficial lavas.

Hrafntinnuhryggur is a rhyolitic fissure exposed within the Krafla volcano, Iceland[28] (Fig. 1). This generated a 2.5 km-long ridge that is parallel to the regional basaltic fissure swarm[29]. The deposits include (1) a feeder dyke exposed at two depths below the surface (i.e. an upper and a lower feeder dyke exposure at ~50 m and ~70 m depth, respectively)[30,31]; (2) lavas comprising obsidian, spherulitic obsidian, and devitrified rhyolite lithofacies, including textural evidence for brecciation and healing/annealing, evidenced by offset flow bands in otherwise coherent samples; and (3) surficial pumice deposits (Fig. 1). The eruption that formed these deposits occurred at ~24 ka in the Last Glacial Period, and lava deposits record interaction with thin ice above the fissure[28]. We attribute the lack of direct evidence for any extensive fall deposits from explosive phase(s) to this inferred ice cover. Here, we explore and re-interpret these deposits in the context of the cryptic fragmentation model, with a focus on both the conduit feeder exposures, and their association with the surface extrusive deposits.

### Textural observations of a silicic conduit

We characterise preserved textures and dissolved volatile concentrations of two transects (from the upper and lower exposures, respectively) across the feeder dyke that is exposed at a depth of ≤70 m beneath the overlying surficial lavas (Fig. 1). The dyke is 3 m wide at its thinnest point. The dyke margin contacts basaltic hyaloclastite country rock[28,31], which abuts a 2–8 cm-thick marginal facies of clast-supported rhyolitic pumice lapilli tuff (pLT). The hyaloclastite is somewhat altered where it contacts the dyke facies; this alteration can result in changes in permeability[32]. Where it is exposed on the western side of the outcrop, the pLT contains hyaloclastite country rock fragments and obsidian pyroclasts, and grades laterally and continuously over <1 cm into dense obsidian (Fig. 1c, d), such that there is no discrete break between the pumice clasts and obsidian (Fig. 2a, b). At distances >1 cm from the margins, dense obsidian then dominates the feeder dyke transects (Fig. 2c–j). At ~110 cm from the western margin of the lower feeder dyke, the black obsidian grades into grey-black, slightly

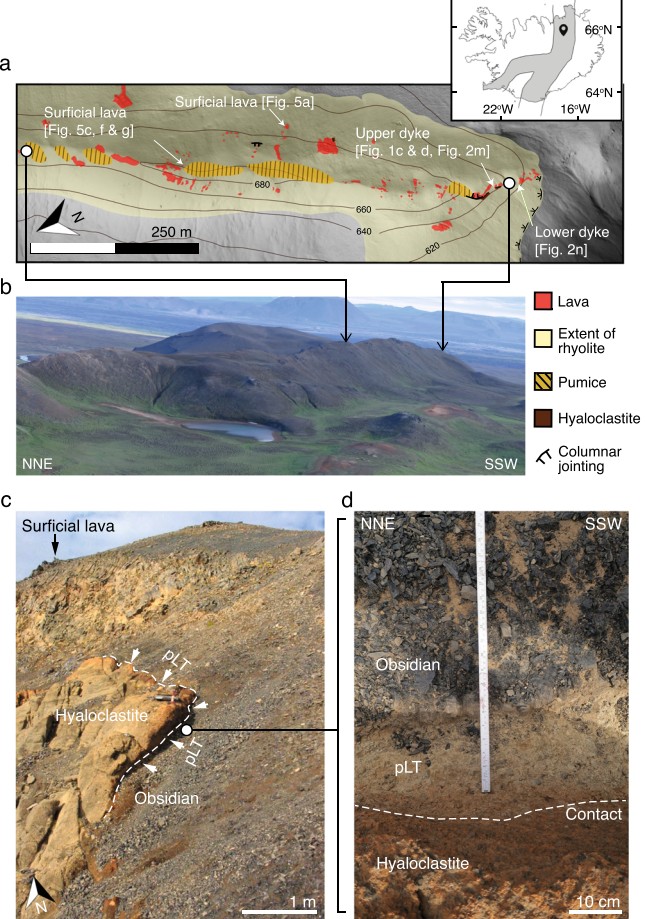

**Fig. 1 | The field localities at Hrafntinnuhryggur. a** A simplified geological map of southern side of the Hrafntinnuhryggur ridgeline showing the locations of lava outcrops, the approximate extent of the intrusive rhyolitic body (demarked by breaks in slope), surficial prominent patches of reworked pumice deposits, cooling joints (see Tuffen & Castro[28]), and the two locations where the hyaloclastite country rock is exposed in contact with rhyolite. The lower and upper feeder dyke exposures are marked as well as the locations of lava-hosted samples presented in Fig. 5. *Inset*: the location of Hrafntinnuhryggur, indicated by the pin symbol, in the Iceland rift zone(s), which are marked in grey. **b** A photograph of the southern part of the ridge (the water body in the foreground is ~45 m in diameter). **c** Field photograph of the upper feeder dyke locality taken looking north, showing the contact between rhyolite and hyaloclastite lined by pumice lapilli-tuff (pLT). **d** A close-up photograph of the contact zone in (**c**). Note that not all lithofacies described at Hrafntinnuhryggur are depicted here (e.g. spheulitic or devitrified rhyolite or rhyolite with columnar jointing); the reader is referred to Tuffen & Castro[28] for more detail.

vesicular obsidian, and then a ~5 cm central portion of the dyke is not preserved/exposed (Fig. 2k, l). From 115 to 130 cm from the western margin, the grey-black vesicular obsidian grades back into black dense obsidian. The grey-black obsidian is distinguished from the black obsidian by its higher relative vesicularity, presence of spherulites, and flow banding. Neither the upper nor the lower feeder dyke outcrops have their eastern margin exposed (Fig. 2k, l).

The obsidian in the feeder dyke exposures is variably spherulitic (Fig. 2e, f and j) but otherwise nominally aphyric. At the western margin of the lower feeder dyke outcrop, where it contacts the pLT, the dyke is consolidated, and there are brown glassy bands which are vesicular and host equigranular millimetre-sized crystalline lithics bearing plagioclase, clinopyroxene and oxides (Fig. 2a). Importantly, the obsidian also contains substantial proportions of hyaloclastite country rock lithic fragments with similar mineralogy and texture to the fine-grained lithics. These distinctively occur within pore spaces, even far from the

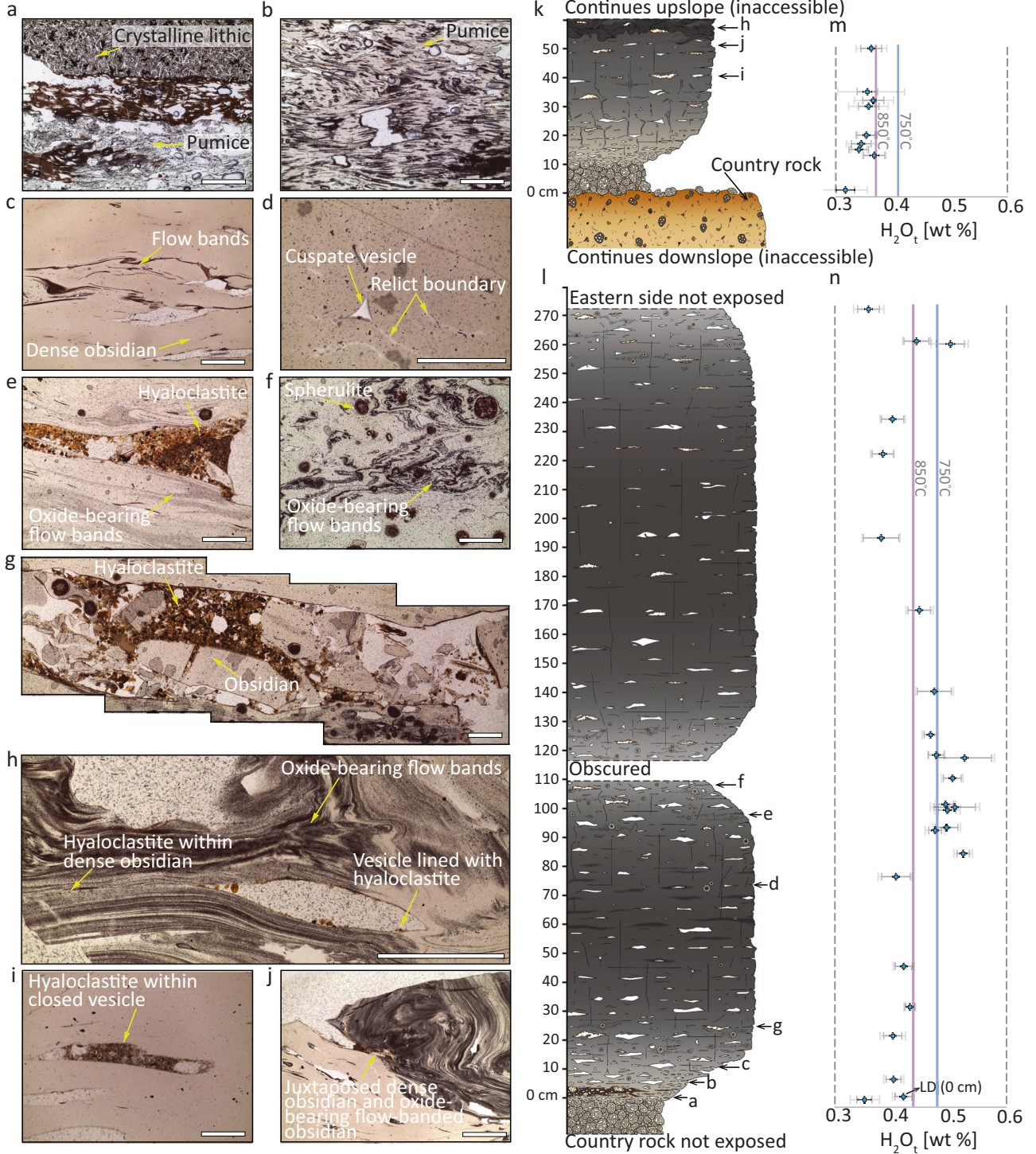

**Fig. 2 | The characteristics of the feeder dyke at Hrafntinnuhryggur.**
**a–j** Photographs of thin sections taken in plane polarized light of the micro-
structures in the upper and lower feeder dyke outcrops (scale bar is 0.5 mm).
**k, l** Logs representing horizontal transects taken from the western margin of the
upper (**k**) and lower (**l**) feeder dyke outcrops. The locations of the thin section
samples in (**a–j**) are indicated. In these logs, the width of the drawn outcrop broadly
reflects the qualitative assessment of induration or, equivalently, degree of
welding[61] (with the exception of the hyaloclastite country rock in (**k**)). **m, n** The total

dissolved $H_2O$ concentration in samples taken from the logs in (**k**) and (**l**), for which
the vertical position scale is the same as in (**k**) and (**l**). We give two sets of error bars:
the lower errors are associated with reproducibility on repeat measurements on a
sample within a 1 cm square ($5 \leq n \leq 25$), and the larger errors are propagated from
the uncertainty on the absorption coefficients (see Methods). The blue and purple
vertical lines represent the solubility of $H_2O_t$ computed for 750 °C and 850 °C,
respectively (see text). See Supplementary Material 1 for additional photo-
micrographs of the thin section samples used in this study.

margin itself and into the conduit core (Fig. 2e and g–j). The hyalo-
clastite lithics are not pervasively incorporated into the groundmass of
the obsidian; they are actually trapped within enclosed vesicle tips
(Fig. 2h). The hyaloclastite lithics appear characteristically orange and

brown in thin section and are all <2.5 mm in size. In the dense obsidian
lithofacies, there are cuspate, convex vesicles which have faint, white
suture lines extending from vesicle tips, and are found at rounded
clast-clast-clast triple junctions (Fig. 2d). In regions of the feeder dyke

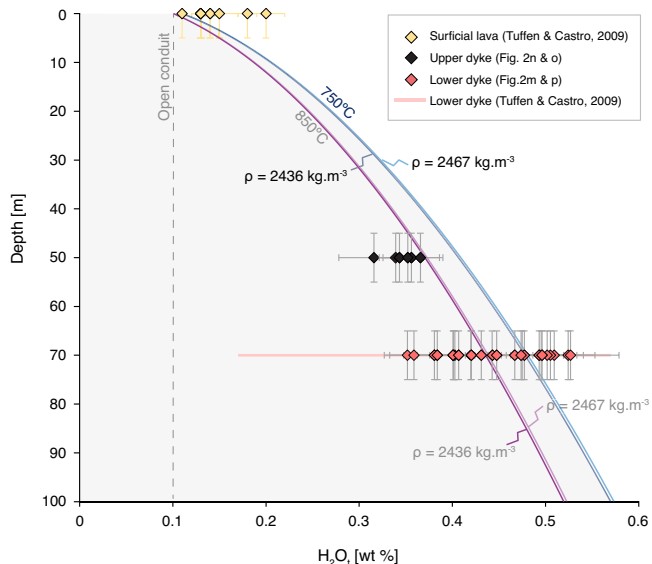

**Fig. 3 | The total $H_2O$ concentration in groundmass glass from obsidian samples from the lower and upper feeder dyke outcrops.** The lower (red) and upper (green) feeder dyke outcrop locations are given in Fig. 1. These $H_2O_t$ values are compared with the $H_2O$ concentrations recorded in the surficial deposits (yellow) from Tuffen & Castro[28]. We also plot the predictions from a rhyolite $H_2O$ solubility model[37]. The pressure is computed assuming magmastatic conditions (see Methods) assuming the magma density $\rho_m$ is $2436 \leq \rho_m \leq 2467$ kg.m$^{-3}$ (measured for these samples) and the temperatures considered at $T = 850\,°C$ (purple curves) and $T = 850\,°C$ (blue curves; see text for details). The vertical dashed line is the prediction for a fully open conduit at atmospheric pressure.

transects that are more vesicular in general (i.e., surrounding the obscured region of the lower feeder dyke exposure and the towards the margins in both the upper and lower feeder dyke exposures) there are few to no triple junctions of clasts, and vesicles are more irregular and elongate (Fig. 2c). Toward the transect centres (i.e. toward the centre of the feeder dyke), wispy features become prominent, defined by dark flow bands with low-angle folding, which pinch and fade out laterally (Fig. 2h–j). In some cases, the domains with dark wispy flow bands are juxtaposed directly with clear rhyolitic glass (Fig. 2j).

We use Fourier transform infra-red spectroscopy (FTIR; see Methods) to determine the total $H_2O$ concentration (termed $H_2O_t$ and determined as wt %) dissolved in the groundmass glass in samples from across the lower and upper feeder dykes. Generalizing these results across both the upper and lower feeder dyke transects, we find that the $H_2O_t$ is relatively lowest at the conduit margins, and highest in the conduit centre (where this is exposed in the lower feeder dyke exposure; Fig. 2m, n). In between the margin and centre, $H_2O_t$ is constant within uncertainty.

### Interpreting $H_2O$ concentration transects

In order to determine whether these $H_2O$ measurements reflect volatile-saturated conditions, we require information about the magmatic temperature. Existing mineral thermometry suggests that rhyolite at Krafla volcano is stored at 850–920 °C[33]. During ascent, fragmentation, and associated adiabatic cooling, this temperature can drop substantially; conservative estimates indicate 750 °C for rhyolitic cases[34]. This is broadly consistent with direct observations of the quenching effect of fragmentation induced by drilling of a rhyolite storage region at Krafla volcano, where fragmental magma quenched in drilling fluid[35]. During this process, volatiles rapidly equilibrated to equivalent temperatures as low as 760 °C[36]. Taking 750–760 °C as the post-fragmentation temperature most appropriate for the shallow accumulation and sintering process at conduit margins, and confining

pressure consistent with depth[28,30], the Liu et al.[37] solubility model for rhyolites would suggest that the lower feeder dyke exposure is either saturated (near the dyke centre; Fig. 2n) or $H_2O$-undersaturated (e.g. away from the dyke centre but not quite at the margins; Fig. 2n). Most of the data for the upper feeder dyke exposure is apparently undersaturated unless the higher emplacement temperature of 850 °C is considered. Crucially, regardless of the temperature assumed, the $H_2O_t$ values vary spatially across the conduits, and at individual horizons, vary such that they cannot all be in equilibrium with an isobaric, magmastatically pressurized column of rhyolite at these depths under isothermal conditions (Fig. 2m, n; Fig. 3).

### The 'cryptic fragmentation' model for lava assembly

In Fig. 4 we show a conceptual model for how the eruption and conduit evolved at Hrafntinnuhryggur, expanding on the existing cryptic fragmentation model[11]. We focus on the details of this model which our new observations contribute.

First, our observation that the dyke edge is lined continuously with poorly/un- sintered pumice clasts in the pLT lithofacies (see Fig. 1d) is consistent with the interpretation that the eruption at Hrafntinnuhryggur began with an explosive phase (Fig. 4a). While the initial explosive phases at other examples of rhyolite eruptions have been sub-Plinian and lasted for days to weeks (such as the eruptions in the Mono-Inyo chain[12,38] or the 2011–12 Cordón Caulle eruption[39] or the 2008 eruption of Chaitén[8]), the eruption at Hrafntinnuhryggur was smaller in total lava volume and fissure length[28]. Therefore, it is difficult to attribute an eruption intensity to the unobservable initial explosive phase. The absence of a subaerial fall deposit, potentially obscured by ice cover[28] during the Last Glacial Period and then post-glacial outwash and deposit reworking, means that this phase cannot be reconstructed further. Nevertheless, the pumiceous conduit lining is consistent with models in which the explosive phase of silicic eruptions involves conduit margin sticking from an early stage[1,2,11–13]. We note that at some sites such as Mule Creek (USA), the extent of shallow country rock damage (in the form of fractures and tuffisites) appears to be far greater than at Hrafntinnuhryggur[22].

Second, we posit that the direct observation of a welding transition from pumice in the pLT to the dense obsidian represents two processes; (1) the thermal insulation of the conduit/dyke during sustained eruption and the deposition of the pLT leading to lower cooling efficiency to the country rock and increased sticking potential, and (2) a change in the particle componentry in deeper magmatic fragmentation below to facilitate welding (see models for welding that implicate pyroclast size as a key parameter where smaller pyroclasts lead to more efficient welding[40]). Previous interpretation of the pumice-obsidian transition at Hrafntinnuhryggur[31] invoked a shift from vanguard fragmental material to the ascent of less-vesicular dyke-filling magma, or to foam collapse, but did not account for magma fragmentation. Here we present unequivocal textural evidence in the dyke proper, including the observation of lithics in the pore spaces, for sinter-assembly (Fig. 2). Furthermore, any model implying wholesale ascent of coherent magma within the dyke interior would require high shear stresses at the conduit wall, at the pLT contact, which would drive pLT deformation and further compaction[41]. The absence of such deformation thus supports the cryptic fragmentation model (Fig. 4b).

These marginal facies—the pLT and into the dense obsidian—are variable in $H_2O_t$ (Figs. 2 and 3) relative to saturation at 750 °C. This variability includes regions that are undersaturated relative to solubility at 750 °C. This is consistent with the $H_2O_t$ from drill core samples in the subsurface feeder system beneath Obsidian Dome[42], California, which are generally at uniform, low (undersaturated relative to magmastatic) $H_2O_t$ regardless of depth[11]. Traditionally, such shallow conduit undersaturation has been explained by invoking the development of a permeable foam through which volatiles can outgas and via which relatively low pressures can be achieved in the gas pore spaces[42]. Here,

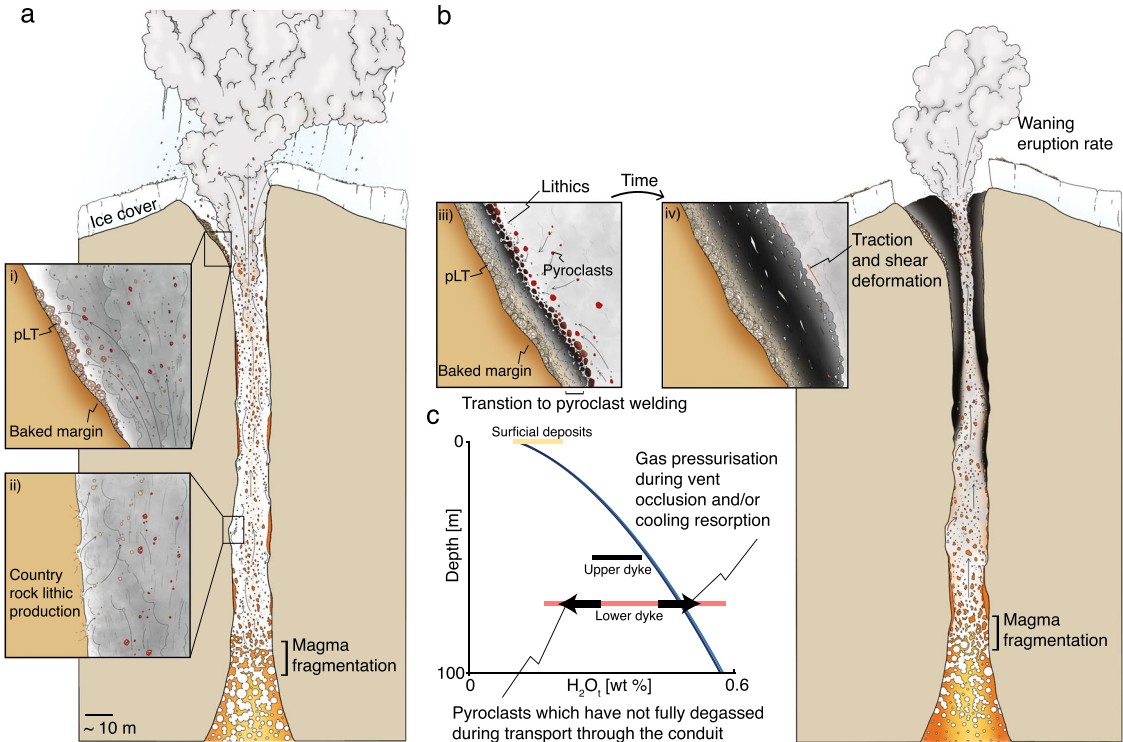

**Fig. 4 | Schematic depiction of the conceptual model for the eruption studied here.** This eruption formed the Hrafntinnuhryggur deposits, which are divided into (**a**) the early purely explosive phase of the eruption and (**b**) the switch to the subsequent hybrid explosive-effusive phase. In both cases, explosive magma fragmentation continues at depth in the conduit, producing a pyroclast dispersion that expands and accelerates up to the Earth's surface. In (**a**) we show visually how pyroclasts can stick to the country rock conduit margins[1,2,11,12,62], forming either welded obsidian patches or pumice lapilli tuff (pLT) with lithics (see *inset i*). *Inset ii* shows how country rock lithics can be dynamically quarried into the dispersion. In (**b**) we show how progressive accumulation of material at the conduit margins leads to a transition from pumice lapilli tuff (pLT) deposition to the formation of welded

dense deposits with lithics incorporated (see *inset iii*). Continued bypassing flow of the pyroclast dispersion during continued explosive eruption can produce tractional shear on the welded deposits, ultimately feeding the effusive portion of the hybrid phase (see *inset iv*). **c** A plot of dissolved $H_2O$ concentration as a function of depth in the conduit (see Fig. 3), showing schematically how the time evolution in our conduit model can explain these data. Importantly, the upper conduit deposits are undersaturated relative to the magmastatically imposed solubility[37], and the relatively lower conduit deposits are initially undersaturated (see conduit margin values in Fig. 2) and can progress toward higher $H_2O$ as the conduit occludes and gas pressure at the base of the sintering window increases (see conduit centre values in Fig. 2).

the textural evidence contains features, such as irregularly shaped vesicles, that could be interpreted as originating from the collapse of bubbles in a magma foam (see Fig. 2c, i). However, these textures can also be explained by the welding of particles, which are likely to be initially pumiceous, to form the obsidian (i.e. akin to pumice fiammé in subaerial ignimbrites). The cryptic fragmentation model has been shown to predict a wide variety of $H_2O_t$ values at the shallow point of particle capture and sintering, dependent on the propensity for the particles to degas diffusively and/or to outgas during transport up the conduit in the pyroclast-gas dispersion[11]. Indeed, Wadsworth et al.[11] demonstrated quantitatively that only pyroclasts with radius $\leq 10^{-5}$ m can degas completely and any others will be captured with elevated $H_2O_t$ values with respect to those in equilibrium with the gas. All of these variations therefore are consistent with the cryptic fragmentation model for shallow rhyolite assembly.

The kind of thorough sintering to a very dense melt body that is inferred here involves the transition from permeable pore spaces between sintering particles to impermeable and isolated pore spaces disconnected from one another[43,44]. The result is therefore that thorough sintering can leave dense obsidian with a small 1–4 vol.% of bubbles/vesicles filled with volatile $H_2O$[45]. On cooling, retrograde solubility[46,47] can then account for the resorption of those final sinter-bubbles to result in and account for non-vesicular obsidian. Similarly, if sintering is occurring in the regime where diffusive equilibrium is relatively slow, then upon final bubble isolation at the end of sintering, diffusive resorption of the trapped $H_2O$ could irradicate the remnant

bubble. However, while this could happen in portions of the feeder dyke studied here, the presence of relict cuspate bubbles suggests that, at least in some areas of the feeder dykes, any cooling was slower than these resorption diffusive processes and slower than the rounding time of the bubbles.

The hyaloclastite country-rock lithic 'dusting' or fragments are frequently found in pore spaces, even far from the margin itself and into the conduit core (Fig. 2). Where they are in pore spaces, they are commonly lining the pore edges (e.g. Fig. 2h). However, they are not pervasively incorporated into the groundmass glass away from the pores. We interpret this via two possible mechanisms. First, the hyaloclastite at Hrafntinnuhryggur has up to 10 wt.% volatile mass (stored in hydrous minerals) that is unstable at >600 °C[32]. Therefore, it is likely that, as lithic fragments are ripped into the pyroclast dispersion from some depth in the conduit and delivered to the shallow sintering window (Fig. 4), they would progressively degas/dewater, liberating gas volume during sintering. As sintering completes to low residual porosity, any addition of gas to the pore spaces will result in pore expansion, and would leave the hyaloclastite particles adhered to the pore edges. A second possibility is that the hyaloclastite particles were intermittently flushed into the sintering assembling lava mass during eruption and only adhered fully to be preserved toward the end of the sintering process when the permeability, and thus the through-going gas flux, dropped to low values[43] sufficient for deposition and sticking. In both cases, the prediction that above fragmentation, the conduit can be under-pressured relative to lithostatic pressure[21] suggests that

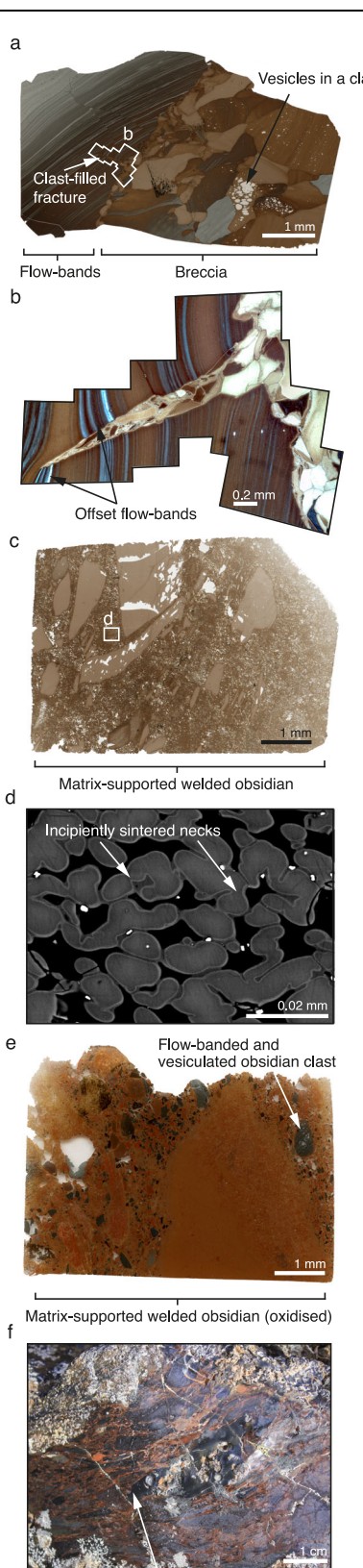

**Fig. 5 | Textural evidence for sintering in lithofacies from the lava deposits exposed at the surface (see Tuffen & Castro[28]). a** A healed breccia (right) juxtaposed with flow banded obsidian containing healed micro-faults (left). **b** Detail of one microfault from (**a**) that is in-filled with obsidian clasts and resembles tuffisite[18]. In (**b**) the deformation of the flow bands appears to record a mixed mode I and mode III (i.e. tensile and tearing) fracture opening mechanism. (c) Incipiently sintered obsidian clasts collected from within a lava outcrop. **d** BSE image of rounded and incipiently sintered clasts from sample (**c**); note that the incipient nature of the sintering is evidenced by the necks labelled in (**d**). **e, f** Oxidised red obsidian with a sintered clastic nature and, in (**f**), for alignment of clasts.

(Fig. 2) can be explained via progressive conduit occlusion and the associated increase in gas pressure in the shallow conduit. As the conduit occludes, the gas pressure is likely to increase, resulting in a higher equilibrium $H_2O_t$ value to which pyroclasts would diffusively degas. A rise in the equilibrium value will slow the diffusive loss of $H_2O_t$ even in disequilibrium during transport. Similar occlusion-related rises in conduit gas pressure have been invoked during hybrid explosive-effusive silicic eruptions to explain lateral gas-fracking into country rock[49] and even the emplacement of laccoliths[1]. If the $H_2O$ is in equilibrium with the gas phase, a rise from ~ 0.4 wt.% to ~ 0.5 wt.% $H_2O_t$ at 750 °C would be associated with an $H_2O$ gas pressure increase of ~ 0.7 MPa (see Methods), which is similar to the overpressure jump inferred by Unwin et al.[49] during conduit welding.

## Direct evidence for the explosive-effusive link

An exceptional feature of the Hrafntinnuhryggur exposure is that the conduit feeder system is in direct stratigraphic conformity with the surficial deposits that it fed. Therefore, this represents a link between the shallow conduit deposits—interpreted above to be assembled from the pyroclastic products of sustained explosive eruption (Fig. 4)—and the overlying, apparently effusive deposits.

The surficial, effusive deposits are mapped and described in detail elsewhere[28], however, here we supplement those descriptions with some key new observations that are pertinent to the explosive-effusive connection. All lithofacies listed are common features along the southern ridge of Hrafntinnuhryggur. The surficial deposits are generally composed of dense lobes on the sides of the top of the ridgeline (Fig. 1) and comprise marginal obsidian and generally more central microcrystalline or spherulitic rhyolite lithofacies. Within the obsidian lithofacies, there exist (1) featureless aphyric and non-vesicular obsidian, (2) flow banded obsidian generally hosting healed fractures that cross cut and offset flow bands by up to a few centimeters[50] (Fig. 5a), (3) clast-supported and thoroughly healed obsidian breccia (Fig. 5a, b) with some breccia clasts internally vesiculated[51] and some invading fractures, (4) matrix-supported welded obsidian fragments (Figs. 5c), and (5) strongly or variably oxidized versions of (4) (Fig. 5e, f). Of these, (2) and (3) tend to occur at the edges of the lava lobes, while (4) and (5) are fracture-hosted features. Located more centrally in the lava lobes are regions of porous, clast-supported obsidian found within fractures. These porous fracture zones are filled with occasional, dense, >1 cm, obsidian clasts which are sometimes flow banded, but primarily comprise small, <1 cm, sub-rounded brown, and dense obsidian particles (Fig. 5). Of the smaller particles, there are rounded, ash-sized obsidian (< 20 microns) which are connected by necks (Fig. 5d).

We interpret the clast supported breccias as forming due to post-sintering late-stage extrusion of the sinter-assembled lava plug. These breccias must have formed above the calorimetric glass transition because they are thoroughly welded. In some cases, the clasts are jigsaw-fit indicating minimal shear strain relative motion, minimal clastic transport, and local brecciation. In other cases, the clast mixture is more polymict and clearly has undergone some local transport before welding and healing (Fig. 5a). Where particles invade fractures in flow banded obsidian (Fig. 5b) the offset and deformation of fracture-adjacent flow bands suggests a mixed mode I and mode II

lithic entrainment in the gas-ash dispersion is possible via moderate conduit implosion events[48]. Regardless of which of these mechanisms is most likely, the pervasive inclusion of lithics in pore spaces cannot be explained by intrusion of the rhyolite without fragmentation.

Finally, in the context of the model introduced above, the elevated $H_2O_t$ values in the central portion of the lower feeder dyke

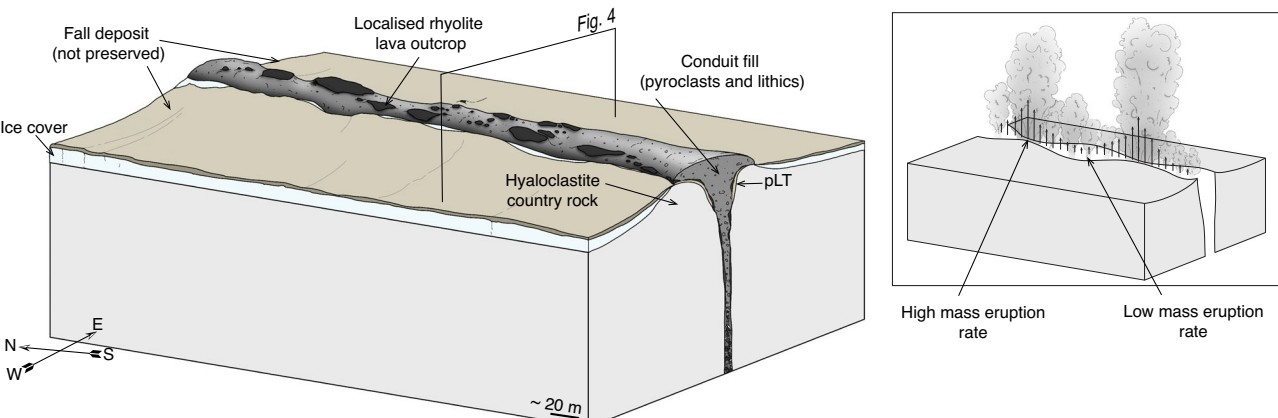

**Fig. 6 | A 3D block diagram of the surface deposits immediately after eruption cessation informed by the present day outcrops (see Fig. 1).** Our model implies that the conduit has upper lava deposits and underlying pyroclastic rubble (see refs. 1,22). The surface lava deposits are discontinuous along strike of the fissure, interspersed by pumice. Any fall deposit from the inferred explosive phase would have been deposited on ice and subsequently reworked during deglaciation. Note that the lateral extent of the lava outcrops and conduit fill as seen from above approximately matches the inferred lateral extent of rhyolite shown in Fig. 1a. *Inset*: the same as in the main figure, but depicted during eruption where we schematically raise the possibility that undulations in fissure width could be responsible for the onset of instabilities in terms of where particle capture and insulated sintering occur most efficiently, compared with higher flux regions where particles may bypass the conduit wall altogether. We propose that this *inset* cartoon can explain lateral variations in lava occurrences in the main panel of the figure.

tensile opening involving some tearing. The lack of matrix-support to the clastic mixture suggests little fluidized transport[18,52] and instead results from the invasion of the locally auto-brecciating obsidian at the lobe margin/base during extrusion.

The matrix supported variably welded material occurs in fractures away from the lava lobe edges and commonly close to the ridge-line axis, such that they are likely to be overlying the conduit itself beneath. These features are consistent with deposition from a fluidized current, and within fractures these are likely to be tuffisites[18,49]. Tuffisites are generally formed by mode I tensile opening, fluidized transport of gas and pyroclasts, and finally fracture closure and sintering. In the context of the cryptic fragmentation model, these features are likely to be late-stage and represent the intermittent gas-fracking of the sinter-assembling plug by the underlying gas-pyroclast mixture that is being replenished by continued magmatic fragmentation[1,2,11]. The presence of these in the surficial deposits is consistent with the proposal from the lower feeder dyke that conduit occlusion can be attributed to gas pressurization, evidenced by an increase in relative $H_2O_t$ in the conduit centre (Fig. 2). Similarly, the variable oxidation of the tuffisitic material in the welded fractures (Fig. 5) is consistent with a fully open conduit in which gas-mixing with air is efficient[2]. Previously, tuffisites have been interpreted as evidence for how magma as a whole degasses[53], whereas here they are a consequence of conduit occlusion by sintering[22].

### 3D conduit geometry and silicic eruption complexity
As described above, geomorphologically, Hrafntinnuhryggur is a ridge with rhyolitic lava outcrops demarking the topmost portion (Fig. 1). Many surficial rhyolite eruptions may begin as fissure eruptions, but rapidly localize into point-vent locations, often with a central cone and/or dome[3]. Nevertheless, it is likely that the sub-surface magma transport is dominantly dyke-fed[54]. Despite evidence for this, most models of conduit ascent dynamics explicitly invoke cylindrical pipe-flow geometries and radial symmetry[11,21,55]. This is perhaps because the extra degree of freedom in dyke flow can introduce substantial additional complexity in the flow field that is solved for. Key to silicic eruptions and the model explored herein, are the consequences of fissure-fed eruption dynamics for the spatial geometry of the fragmentation front, and/or where in the upper conduit sintering occurs. While there exist no quantitative predictions of this kind of silicic eruption via which to address these issues, we use the observation that

the surficial lava lobes are spatially separated by extensive patches of pumice on the Hrafntinnuhryggur ridgeline (Fig. 1a) to explore this in 3D (Fig. 6). In Fig. 6, we depict a simplified 3D block diagram of the ridge in order to emphasize that the dense rhyolite lava outcrops are discontinuous, separated by patches of loose, poorly exposed, and dominantly unconsolidated material. It is possible that localization from fissure- to central-vent geometry at the surface could be controlled top-down by the development of loci of sintering along-strike. Similarly, if at depth, fragmentation does not occur at the same explosivity all along the strike of the fissure, then the flow regimes may favour sintering first in some locations, whereas in other locations along strike, the pyroclastic flow may simply bypass the margin and not deposit. The spatial organisation of these processes will evolve in time during the eruption. These considerations, underpinned by the field evidence at Hrafntinnuhryggur, open up the possibility that silicic eruption dynamics may be substantially more complex and spatially coupled than has yet been explored.

## Methods
### Thin sectioning and microscopy
In-situ samples were collected to a high spatial resolution (samples taken approximately every 5–30 cm) of the lower and upper conduit transect. Polished thin sections were made of all samples collected across the conduit transects and surficial lava lithofacies that were of interest. The thin sections were examined under an optical microscope. A subset of samples was examined using back-scatted electron (BSE) images taken using a Hitachi SU-70 field emission scanning electron microscope at Durham University, with a 15 kV beam voltage and a 15 mm working distance.

### Fourier-transform infra-red spectroscopy (FTIR)
We use FTIR to determine the total $H_2O$ concentration in obsidian samples (see Supplementary Data 1 and 2). From the hand samples, millimetric sub-samples were doubly polished by hand using SiC papers to create wafers of thickness 100–250 μm. Using transmitted light optical microscopy, the wafers were inspected to check that they were glassy and to identify regions of optically clear glass devoid of vesicles or rare microlites. We used a Thermo-Nicolet infra-red spectrometer with a Continuum Analytical microscope, KBr beamsplitter, and a MCT-A detector. Spectra were collected across wavenumbers 4000 to 1000 cm$^{-1}$ at a resolution of 4 cm$^{-1}$. Raw spectra were

processed using an 10–12-point baseline and the height to peaks in the spectra were measured from that baseline (Fig. 1 in Supplementary Material 2). Peaks were identified at 3550, and 1630 cm$^{-1}$ wavenumbers, consistent with total $H_2O$, termed $H_2O_t$, and molecular $H_2O$, termed $H_2O_m$, respectively[56]. No $CO_2$ peak at 2530 cm$^{-1}$ was found in these samples. The concentration of a given species was found by using the Beer-Lambert law $C_i = M_i H/(d\rho\epsilon)$, where subscript $i$ refers to the species of interest (e.g. $H_2O_t$), $M_i$ is the species molecular weight (e.g. 18.02 g.mol$^{-1}$ for $H_2O$), $H$ is the measured peak height above baseline, $d$ is sample thickness, $\rho$ is sample density, and $\epsilon$ is the absorptivity coefficient. In practice, we use the McIntosh et al.[57] method to account for the species-dependence of $\epsilon$, which involves (1) measurement of $H$ at 1630 cm$^{-1}$ and use of $\epsilon_{1630} = 55 \pm 2$ l.mol$^{-1}$.cm$^{-1}$ to give $C_{H_2O_m}$, (2) measurement of $H$ at 3550 cm$^{-1}$ and using

$$C_{OH} = \frac{1}{\epsilon_{3500[OH]}}\left(\frac{M_i H}{d\rho} - \epsilon_{3500[H_2O_m]}C_{H_2O_m}\right) \qquad (1)$$

to find $C_{OH}$ with $\epsilon_{3500[OH]} = 100 \pm 2$ l mol$^{-1}$.cm$^{-1}$ and $\epsilon_{3500[H_2O_m]} = 56 \pm 4$ l mol$^{-1}$.cm$^{-1}$, and (3) assuming $C_{H_2O_t} = C_{H_2O_m} + C_{OH}$. During the measurements, an aperture of $100 \times 100$ μm was used. The sample thickness was measured directly at each spot location using one of two methods. First, measurements were made using a profilometer accurate to 1 μm across the wafer. In the region of a given FTIR measurement, up to 10 profilometer measurements were taken and averaged. Second, FTIR was re-run in reflectance mode where reflectance spectra show $m$ number of fringes between 2400 and 2800 cm$^{-1}$, which allowed thickness to be computed using $d = m(2n\Delta w)^{-1}$ where $n = 1.5$ is the refractive index and $\Delta w$ is the difference in wave number between the two limits where fringes were observed (i.e. $\Delta w = 400$ cm$^{-1}$ in this example)[56]. The $d$ measured by profilometry agrees with that measured by fringe counting with a coefficient of determination of $r^2 = 0.994$ when an intercept of 0 is assumed (see Supplementary Material 2 and Supplementary Data 2). Sample density was determined by using a density calculator for volcanic glass[58] with published Hrafntinnuhryggur glass compositions[28,59] and assuming ambient laboratory temperature. To view spectra locations on the obsidian wafers used with FTIR, see Supplementary Material 3.

## Uncertainty in $H_2O$ determination

Using FTIR there is uncertainty on each of the parameters in the Beer-Lambert law. To account for this, we took the maximum and minimum values on each of $\epsilon$, $\rho$, and $d$ to compute maximum and minimum possible $C_i$ values for each species. The uncertainties on the respective $\epsilon$ values are quoted above. The uncertainty on $d$ arises from the standard deviation on repeat measurements using the profilometer around the location of a given spot FTIR measurement and are, on average, $\pm 2.5$ μm (exceeding the measurement uncertainty). The uncertainty on $\rho$ arises from the variation in the glass composition used in the density calculator[28,58,59] and is, on average $\pm 15.5$ kg.m$^{-3}$. By propagating the respective uncertainties in this manner, coupled with the species-dependent $\epsilon$ values using Mcintosh et al.[57], we can accurately report the total uncertainties on our $H_2O_t$ determinations, which always exceed the typical reported analytical uncertainty associated with $H$ (e.g. see Figs. 2 and 3).

## Pressurization associated with conduit occlusion

We use the Liu et al.[37] solubility model, which is calibrated for rhyolitic melts and takes inputs of $H_2O$ pressure and temperature. Using this model, the results in Fig. 2 demonstrate that the approximate value of $H_2O_t$ in much of the feeder dyke away from the dyke core is ~0.4 wt.%. In the dyke core, this approximate value is ~0.4–0.5 wt.%. At 750 °C, 0.4 wt.% is equilibrium at 1.26 MPa and 0.5 wt.% is equilibrium at 1.96 MPa. Therefore, to explain the increase of $H_2O$ observed in the dyke core, the pyroclast capture and sintering environment during

accumulation of the dyke core material would be accompanied by a pressure rise of +0.7 MPa. This value is broadly consistent with pressure changes inferred elsewhere[9,17,60] albeit by different mechanisms in the context of different eruption models than those invoked here.

## Data availability

All data pertaining to these results are given in the Supplementary Material 1, 2 and 3, and Supplementary Data 1 and 2.

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

## Acknowledgements

Thanks to Ed Llewellin and Jim Gardner for their role in developing the cryptic fragmentation ideas[11] extended herein. Additional thanks to David Zhang, Alexandra Kushnir, Lucille Carbillet, Taylor Witcher, Dave McGarvie, Jon Castro, Jamie Farquharson, Jérémie Vasseur, and C. Ian Schipper. Thanks to Elodie Saubin, Ben Kennedy, Kim Berlo, Ellen McGowan, Jackie Kendrick, Yan Lavallée, and Bettina Scheu for stimulating discussions in the field. We thank Landsvirkjún and the staff at the Krafla Geothermal Power Plant for their hospitality during fieldwork. Funding was provided by PhD studentships to A. Foster via the IAPETUS Doctoral Training Program (grant number NE/S007431/1) and to H. Unwin via the ENVISION Doctoral Training Program and a BUFI grant from the British Geological Survey. H. Tuffen was supported by a Royal Society Fellowship. We thank Leon Bowen for assistance with scanning electron microscopy at Durham University, and Ian Chaplin, Sophie Edwards and Samantha Thorpe for assistance with wafer preparation.

## Author contributions

A. Foster and F.B. Wadsworth conceived this study and A. Foster collected the textural and $H_2O$ concentration data. H. Tuffen assisted with $H_2O$ measurements and co-conceived this study. H. Tuffen and H. Unwin provided fieldwork support. M. Humphreys provided supervision support throughout. All authors contributed to the manuscript.

## Competing interests

The authors declare no competing interests.
