## [Peer Review File · Nature Communications]

REVIEWER COMMENTS

Reviewer #1 (Remarks to the Author):

This paper provided detailed field observations and water content profiles for the conduit and surficial lava at Hrafninnuhryggur. The sintering of fragmented pyroclasts and advection may play an important role in the transition of eruption styles. Hrafninnuhryggur, Krafla volcano, Iceland is a rhyolitic fissure with the exposure of conduit. Based on the lava textures suggesting the fragmentation and the low water content in the conduit, the authors concluded that effusive eruption of sintered pyroclasts. The quantitative analysis of the water content profile at the edge of the conduit and surficial lava simultaneously is rare and informative. The manuscript is well written. One thing I noted is that the consistency of Figs 1a, 4b, and 6 is not clear to me.

In Fig.4b, the authors provide the image that explosive eruption continues at the center of the conduit while the lava close to the wall is sintered. I am not sure whether both processes can occur simultaneously in the EW conduit cross-section if we consider the variation in the strike of the fissure direction (NS). This is because, in Fig.1a, there is a separation between the pumice and lava areas at the fisher axis. It looks like some part of the fissure was clogged, and the explosive activity was localized. Actually, in line 289, the authors mention the localization effect. In contrast, Fig.6 appears to show that after the pumice-generating explosive eruption ended, some localized lava flow reached the surface. Please check consistency.

I also noted that the relation between the “extent of rhyolite” in Fig.1a and Fig.6 is unclear.

Minor issues:

Line 149: Is this because the exposure is too short?

Line 238: The authors mention “lobes,” but it is difficult to identify lobes in Fig.1c. It may be informative if Fig.5 includes the large-scale picture of surficial lava.

Line 244: It is not clear what “matrix” means.

Line 321: McIntosh

Fig1a inset: I cannot find the pin symbol showing the location of Hrafninnuhryggur.

Fig1a: Regarding the lava symbol, what is the difference between red and pink? The legend defines Hyaloclastite as brown, but it is difficult to distinguish the brown color on the map.

Fig.2a-j: The notes are too small and hard to read.

Fig.3 legend: surfical

Reviewer #2 (Remarks to the Author):

Dear Authors

Please find my review in the attached document, for minor comments and figure editing see the

annotated pdf.

Review on Foster et al 2024, Evidence for the formation of silicic lava by sintering recorded in situ in a dissected feeder dyke

Understanding the densification of rhyolitic magma to produce degassed, low porosity melt for effusive eruption is a major challenge in volcanology. After the permeable foam model (1986) there were several explanations but the 2008, Chaitén and 2011-12 eruptions of Cordón Caulle really changed our understandings.

The Hrafninnuhryggur obsidian has been extensively studied in volcanology, including physical volcanology (Tuffen & Castro 2009, Saubin et al 2019), formation of spherulites (Castro et al. 2008,2009) and retrograde solubility of H₂O (Ryan et al 2015).

This study of Hrafninnuhryggur volcano is a new contribution to broaden the knowledge of natural sintering providing insights on hybrid (explosive-effusive eruptions) and the formation of the dense, obsidian conduit rocks. The investigation methodology is sound. The sampling resolution (5-30cm) is sufficient to reveal the conduit transects in detail. Excellent thin sectioning and microscopy document all textural details for the sintered origin (cusped relict boundaries, country rock lithics). The FTIR measured H₂O data well describes the vertical and lateral dyke variations. Using Liu et al. solubility model, the saturated (dyke center) and undersaturated conditions (margins) could be interpreted by overpressure jumps during conduit welding.

The real significance of this study gives is the link between explosive and effusive activity. The article uses the “cryptic fragmentation model” that was framed on the Chaitén (2008) and Cordón Caulle (2011-2012) eruptions (Wadsworth et al 2020,2022). The initial explosive phase material was not preserved at the surface and can only be sampled from the dyke wall. However, there was significant sintering in the overlying lavas. The provided textural data well support the hypothesis of the in-conduit explosive fragmentation and the re-aggregation of the clasts via sintering.

These processes have only been documented for recent eruptions or young occurrences. Based on detailed reconstructions from such field data and experimentally produced textures (Gardner et al. 2019) such processes are expected to be recognised in older volcanic formations, where volcanic vents are erosionally exposed (eg. Mule Creek).

The high-quality figures support the sample description and model illustrations well. The size of the sample photos in Figure 2 is quite small, but they are included in the supplementary.

I recommend that the paper be published after a moderate revision. In addition, some figure editing will help to improve the layout. My questions are listed here, for minor comments see the annotated pdf.

Questions:

- Sintering/welding intensity can be supported by quantitative textural data (porosity, clast elongation) and can be ranked as for welding (eg. Quane & Russel 2005)?
- You used 750-760 °C interval for post fragmentation temperature which is appropriate for the shallow accumulation and sintering process at conduit margins. If we use the glass transition terminology, is it possible to determine some temperature difference on the dyke transect based on the presence of spherulites (above T_g) and breccias (below T_g)?

- Ryan et al 2015 published the retrograde solubility model. Is the degassing quench or re-hydration quench relevant here since the sample material was the same.

-Intrusive processes have both mechanical and thermal impacts on surrounding country rocks. Saubin et al. 2019 stated that the basaltic hyaloclastite at the contact is discoloured over <30 cm and fractured, with a decrease in matrix permeability by >1 order of magnitude. What was the expected temperature of the host rock alteration around the dyke?

- Tuffen & Castro 2009 documented perlitic alteration in the effusive facies. In terms of H₂O content was there any interaction between the magma and the host rock in the dyke?

- Sintering is grain size dependent, and effective sintering requires grains size fractionation. Is there an explanation/model in fragmentation processes for why the fine material remains in the conduit during a sustained explosive eruption?

Reviewer comments are given in full in grey. Author replies are given in black. Any changes to the manuscript are indicated in blue for the editor's convenience.

Reviewer 1

This paper provided detailed field observations and water content profiles for the conduit and surficial lava at Hrafninnuhryggur. The sintering of fragmented pyroclasts and advection may play an important role in the transition of eruption styles. Hrafninnuhryggur, Krafla volcano, Iceland is a rhyolitic fissure with the exposure of conduit. Based on the lava textures suggesting the fragmentation and the low water content in the conduit, the authors concluded that effusive eruption of sintered pyroclasts. The quantitative analysis of the water content profile at the edge of the conduit and surficial lava simultaneously is rare and informative. The manuscript is well written.

One thing I noted is that the consistency of Figs 1a, 4b, and 6 is not clear to me. In Fig.4b, the authors provide the image that explosive eruption continues at the center of the conduit while the lava close to the wall is sintered. I am not sure whether both processes can occur simultaneously in the EW conduit cross-section if we consider the variation in the strike of the fissure direction (NS). This is because, in Fig.1a, there is a separation between the pumice and lava areas at the fisher axis. It looks like some part of the fissure was clogged, and the explosive activity was localized. Actually, in line 289, the authors mention the localization effect. In contrast, Fig.6 appears to show that after the pumice-generating explosive eruption ended, some localized lava flow reached the surface. Please check consistency.

This is a really interesting point. We now have substantially expanded our discussion of the localisation phenomena. First and foremost, we now include an *inset* to **Fig. 6** that depicts some of the possible along-strike variations in mass eruption rate and sinterability. That new figure is copied here:

Similarly in the final section we now discuss these phenomena more completely with the hope that readers such as this Reviewer can see how we envisage along-strike effects playing a key role in localisation.

I also noted that the relation between the “extent of rhyolite” in Fig.1a and Fig.6 is unclear.

The ‘extent of rhyolite’ in **Fig. 1a** is determined by breaks in slope (as described in the caption). By contrast, in **Fig. 6** we document the

Note that the lateral extent of the lava outcrops and conduit fill as seen from above approximately matches the inferred lateral extent of rhyolite shown in **Fig. 1a**.

Line 149: Is this because the exposure is too short?

Line 149 in the submitted version is associated with the H₂O profile variations in the feeder dykes. We infer that the Reviewer is asking if we suspect that there would be an H₂O-enriched dyke core in the upper feeder dyke, just as there is in the lower feeder dyke, if there was more complete exposure. If this is the question being asked, then yes, we believe that would be a reasonable expectation. However, we do not have sufficient direct evidence for that and so we do not add speculation along these lines in the manuscript. However, it has been an interesting discussion raised by this question.

Line 238: The authors mention “lobes,” but it is difficult to identify lobes in Fig.1c. It may be informative if Fig.5 includes the large-scale picture of surficial lava.

The lobe morphology of the lava is part of the physical volcanology described in Tuffen & Castro (2009) and they are depicted schematically in **Fig. 6**.

Line 244: It is not clear what “matrix” means.

‘Matrix supported’ is a standard term used in the description of pyroclastic rocks. **Fig. 5c** comprises some large obsidian clasts in a finer-grained matrix. The large obsidian clasts do not commonly touch one another and so the fine-grained matrix is supporting the clasts. We now remove reference to **Fig. 5d** associated with the phrase ‘matrix supported’ because this was incorrect. We thank the Reviewer for pointing this out.

Line 321: McIntosh

Corrected.

Fig1a inset: I cannot find the pin symbol showing the location of Hrafninnuhryggur. Fig1a: Regarding the lava symbol, what is the difference between red and pink? The legend defines Hyaloclastite as brown, but it is difficult to distinguish the brown color on the map.

This was used to denote the difference between direct exposure (red) and inferred exposure (pink), as is common in geological mapping. However, we acknowledge that this could be confusing and so now we use red exclusively here and only mark the direct exposure. The pin symbol is now added.

Fig.2a-j: The notes are too small and hard to read.

We have now increased the size of the font in the notes to aid legibility. We note that when this figure is printed at full page width it will be large by approximately 15-20%.

Fig.3 legend: surfical

This is now fixed and reads “surficial” as it should. We thank the Reviewer for catching this.

Reviewer 2

Understanding the densification of rhyolitic magma to produce degassed, low porosity melt for effusive eruption is a major challenge in volcanology. After the permeable foam model (1986) there were several explanations but the 2008, Chaitén and 2011-12 eruptions of Cordón Caulle really changed our understandings. The Hrafninnuhryggur obsidian has been extensively studied in volcanology, including physical volcanology (Tuffen & Castro 2009, Saubin et al

2019), formation of spherulites (Castro et al. 2008,2009) and retrograde solubility of H₂O (Ryan et al 2015).

This is an accurate summary. We would add that the Hrafninnuhryggur obsidian has been additionally used to understand hydrous welding dynamics (Wadsworth et al. 2019, 2021), viscoelastic magma rupture physics (Tuffen et al. 2008; Wadsworth et al. 2018), and gas-ash reaction rates (Casas et al. 2019), indeed making it a well-studied material. However, the emplacement of the obsidian and associated physical volcanology is less well understood (Tuffen & Castro 2009).

This study of Hrafninnuhryggur volcano is a new contribution to broaden the knowledge of natural sintering providing insights on hybrid (explosive-effusive eruptions) and the formation of the dense, obsidian conduit rocks. The investigation methodology is sound. The sampling resolution (5-30cm) is sufficient to reveal the conduit transects in detail. Excellent thin sectioning and microscopy document all textural details for the sintered origin (cusped relict boundaries, country rock lithics). The FTIR measured H₂O data well describes the vertical and lateral dyke variations. Using Liu et al. solubility model, the saturated (dyke center) and undersaturated conditions (margins) could be interpreted by overpressure jumps during conduit welding.

The real significance of this study gives is the link between explosive and effusive activity. The article uses the “cryptic fragmentation model“ that was framed on the Chaitén (2008) and Cordón Caulle (2011-2012) eruptions (Wadsworth et al 2020,2022). The initial explosive phase material was not preserved at the surface and can only be sampled from the dyke wall. However, there was significant sintering in the overlying lavas. The provided textural data well support the hypothesis of the in-conduit explosive fragmentation and the re-aggregation of the clasts via sintering. These processes have only been documented for recent eruptions or young occurrences. Based on detailed reconstructions from such field data and experimentally produced textures (Gardner et al. 2019) such processes are expected to be recognised in older volcanic formations, where volcanic vents are erosionally exposed (eg. Mule Creek). The high-quality figures support the sample description and model illustrations well. The size of the sample photos in Figure 2 is quite small, but they are included in the supplementary. I recommend that the paper be published after a moderate revision. In addition, some figure editing will help to improve the layout. My questions are listed here, for minor comments see the annotated pdf.

We are grateful for this endorsement of our work and associated methods.

- Sintering/welding intensity can be supported by quantitative textural data (porosity, clast elongation) and can be ranked as for welding (eg. Quane & Russel 2005)?

This is a valuable suggestion from this Reviewer. We have taken this on board and now provide a reference to Quane & Russell (2005) in the caption to Figure 2 where we suggest that the lateral extent of the deposits depicted in Figures 2n and 2m relate to “welding degree”.

- You used 750-760 °C interval for post fragmentation temperature which is appropriate for the shallow accumulation and sintering process at conduit margins. If we use the glass transition terminology, is it possible to determine some temperature difference on the dyke transect based on the presence of spherulites (above T_g) and breccias (below T_g)?

This is not possible with the information we have. That is because the formation/growth of spherulites (which are sparse in the transects; **Fig. 2**) depend so much on the initial H₂O concentration and, for iron-rich rhyolite like Hrafninnuhryggur, also the initial iron oxidation state (Castro et al. 2009).

Neither of these parameters are known *prior* to spherulite formation and use of them as temperature indicators would require a forensic investigation of diffusion gradients around spherulites, which we have not attempted. Similarly, any brecciation is not dictated solely by a temperature, but also by a local strain rate. While we have evidence for shear strain (our Figure 2 & depicted in Figure 4 as “tractional shear”), the evidence in the field is insufficient to use this to constrain a strain rate. Indeed, the traction-derived shear dynamics depicted schematically in **Fig. 4** are poorly investigated altogether and so it is hard, if not impossible, presently, to infer rates and temperatures from these textures. We note however, that these are interesting suggestions from the Reviewer.

- Ryan et al 2015 published the retrograde solubility model. Is the degassing quench or rehydration quench relevant here since the sample material was the same.

This comment is an interesting one. We have now added mention of retrograde solubility and associate resorption as a mechanism to ‘remove’ the final gas volume fraction and to convert sintered obsidian to genuinely fully-dense obsidian. That new text reads as follows:

The kind of thorough sintering to a very dense melt body that is inferred here involves the transition from permeable pore spaces between sintering particles to impermeable and isolated pore spaces disconnected from one another^{43,44}. The result is therefore that thorough sintering can leave dense obsidian with a small 1-4 vol.% of bubbles/vesicles filled with volatile H₂O⁴⁵. On cooling, retrograde solubility^{46,47} can then account for the resorption of those final sinter-bubbles to result in and account for non-vesicular obsidian. Similarly, if sintering is occurring in the regime where diffusive equilibrium is relatively slow, then upon final bubble isolation at the end of sintering, diffusive resorption of the trapped H₂O could irradiate the remnant bubble. However, while this could happen in portions of the feeder dyke studied here, the presence of relict cusped bubbles suggests that, at least in some areas of the feeder dykes, any cooling was slower than these resorption diffusive processes and slower than the rounding time of the bubbles.

-Intrusive processes have both mechanical and thermal impacts on surrounding country rocks. Saubin et al. 2019 stated that the basaltic hyaloclastite at the contact is discoloured over <30 cm and fractured, with a decrease in matrix permeability by >1 order of magnitude. What was the expected temperature of the host rock alteration around the dyke?

We have no way to determine this using the data we have collected. Weaver et al. (2020) discuss the hyaloclastite alteration in some detail. They show that heating of hyaloclastite can result in permeability increases as well as decreases, depending on the confining pressure regime. In their paper on the Krafla hyaloclastite itself, Weaver et al. (2020) do not conclude anything specific or useful here about the Hyaloclastite behaviour in response to the rhyolite eruption. However, on the basis of this comment we now cite that paper so that the interested reader can understand the alteration around the feeder dyke(s). We make this new citation in the section *Primary observations of a silicic conduit*:

The hyaloclastite is somewhat altered where it contacts the dyke facies; this alteration can result in changes in permeability³²

- Tuffen & Castro 2009 documented perlitic alteration in the effusive facies. In terms of H₂O content was there any interaction between the magma and the host rock in the dyke?

We find no evidence for such interaction recorded in the H₂O concentrations we measure. However, there is clear evidence for host-rock interaction recorded by the hyaloclastite clasts/debris found within the sealed pore spaces of the feeder dyke(s); see **Fig. 2**.

- Sintering is grain size dependent, and effective sintering requires grains size fractionation. Is there an explanation/model in fragmentation processes for why the fine material remains in the conduit during a sustained explosive eruption?

This question is interesting and is studied extensively by Farquharson et al. (2022), which is cited and discussed in our manuscript too. We now add to that discussion as follows:

We have made almost every change suggested by the Reviewer on the pdf manuscript directly. Here we describe the few places where we were unable to do so:

- The upper and lower feeder dyke exposures are difficult to photograph from a distance (see Fig. 1c) because of the steep slope on which they are exposed. For this reason we are not able to convincingly indicate the exact transect locations on Fig. 1, as requested.
- Where we place 0 cm on the transects in Figs 2m and 2n is arbitrary and so we have not made a change to place both at the base of the pLT. Additionally, the base of the pLT in Fig. 2n is not exposed.
- We have opted not to differentiate “internal” from “external” tuffisites, because the tuffisitic material’s relationship with the country rock is not always clear at the Hrafninnuhryggur site. This is because the country rock is only exposed in contact with the rhyolite in a few sparse places.

References cited in these replies

- Casas, A.S., Wadsworth, F.B., Ayris, P.M., Delmelle, P., Vasseur, J., Cimarelli, C. and Dingwell, D.B., 2019. SO₂ scrubbing during percolation through rhyolitic volcanic domes. *Geochimica et Cosmochimica Acta*, 257, pp.150-162.
- Castro, J.M., Cottrell, E., Tuffen, H., Logan, A.V. and Kelley, K.A., 2009. Spherulite crystallization induces Fe-redox redistribution in silicic melt. *Chemical Geology*, 268(3-4), pp.272-280.
- Quane, S.L. and Russell, J.K., 2005. Ranking welding intensity in pyroclastic deposits. *Bulletin of Volcanology*, 67, pp.129-143.
- Tuffen, H., Smith, R. and Sammonds, P.R., 2008. Evidence for seismogenic fracture of silicic magma. *Nature*, 453(7194), pp.511-514.
- Tuffen, H. and Castro, J.M., 2009. The emplacement of an obsidian dyke through thin ice: Hrafninnuhryggur, Krafla Iceland. *Journal of Volcanology and Geothermal Research*, 185(4), pp.352-366.
- Wadsworth, F.B., Witcher, T., Vossen, C.E., Hess, K.U., Unwin, H.E., Scheu, B., Castro, J.M. and Dingwell, D.B., 2018. Combined effusive-explosive silicic volcanism straddles the multiphase viscous-to-brittle transition. *Nature communications*, 9(1), p.4696.
- Wadsworth, F.B., Vasseur, J., Schaubroth, J., Llewellyn, E.W., Dobson, K.J., Havard, T., Scheu, B., von Aulock, F.W., Gardner, J.E., Dingwell, D.B. and Hess, K.U., 2019. A general model for welding of ash particles in volcanic systems validated using in situ X-ray tomography. *Earth and Planetary Science Letters*, 525, p.115726.
- Wadsworth, F.B., Vasseur, J., Llewellyn, E.W., Brown, R.J., Tuffen, H., Gardner, J.E., Kendrick, J.E., Lavallée, Y., Dobson, K.J., Heap, M.J. and Dingwell, D.B., 2021. A model for permeability evolution during volcanic welding. *Journal of Volcanology and Geothermal Research*, 409, p.107118.

REVIEWERS' COMMENTS

Reviewer #1 (Remarks to the Author):

The authors addressed my comments. I recommend publication.

The following are just comments. I appreciate that the authors added the inset figure in Fig.6. This will help readers. In the inset, the authors illustrate the high mass eruption rate at the narrow regions. I infer the relation between the mass eruption rate and the width of the fissure is complex and depends on the boundary conditions. Especially in this case, the width of the fissure is subject to change by clogging. Vigorous eruptions may keep the broader vent. Indeed, the authors denote a more explosive phase in the broader fissure in Fig. 4. Thus, I consider denoting the width dependence in the inset figure unnecessary. However, since the manuscript has already been revised and this is not an essential point of this paper, it is up to the authors to modify this inset.

Reviewer #2 (Remarks to the Author):

Dear Authors,

Thank you for your comprehensive responses to the review. After reading the revised version I have no further comments.

Kind regards.